# Tracing the Lineage of Two Traits Associated with the Coat Protein of the *Tombusviridae*: Silencing Suppression and HR Elicitation in *Nicotiana* Species

**DOI:** 10.3390/v11070588

**Published:** 2019-06-28

**Authors:** Mustafa Adhab, Carlos Angel, Andres Rodriguez, Mohammad Fereidouni, Lóránt Király, Kay Scheets, James E. Schoelz

**Affiliations:** 1Division of Plant Sciences, University of Missouri, Columbia, MO 65211, USA; 2Department of Plant Protection, University of Baghdad, 10071 Baghdad, Iraq; 3National Coffee Research Center-Cenicafe, Planalto, km. 4, Vía antigua Chinchiná-Manizales, Manizales (Caldes), Colombia; 4Department of Pathophysiology, Plant Protection Institute, Centre for Agricultural Research, Hungarian Academy of Sciences, H-1022 Budapest, Herman Ottó str. 15, Hungary; 5Department of Plant Biology, Ecology, and Evolution, Oklahoma State University, Stillwater, OK 74078, USA

**Keywords:** virus effectors, host resistance, hypersensitive response, virus silencing suppressors, avirulence, tombusvirids

## Abstract

In this paper we have characterized the lineage of two traits associated with the coat proteins (CPs) of the tombusvirids: Silencing suppression and HR elicitation in *Nicotiana* species. We considered that the tombusvirid CPs might collectively be considered an effector, with the CP of each CP-encoding species comprising a structural variant within the family. Thus, a phylogenetic analysis of the CP could provide insight into the evolution of a pathogen effector. The phylogeny of the CP of tombusvirids indicated that CP representatives of the family could be divided into four clades. In two separate clades the CP triggered a hypersensitive response (HR) in *Nicotiana* species of section *Alatae* but did not have silencing suppressor activity. In a third clade the CP had a silencing suppressor activity but did not have the capacity to trigger HR in *Nicotiana* species. In the fourth clade, the CP did not carry either function. Our analysis illustrates how structural changes that likely occurred in the CP effector of progenitors of the current genera led to either silencing suppressor activity, HR elicitation in select *Nicotiana* species, or neither trait.

## 1. Introduction

Plant defenses against viral pathogens may be divided into two broad categories. The first line of defense in plants involves their capacity to recognize viral nucleic acids, typically double stranded RNAs, and to mobilize the gene silencing machinery to target the viral RNAs for degradation [1,2]. This defense response is known as RNA silencing (or RNAi), and several recent reviews have noted the similarity of the RNAi defense against viruses to Pathogen-Associated Molecular Pattern (PAMP)-Triggered Immunity (PTI), which has been characterized for bacterial, fungal and oomycete pathogens [3,4,5]. Indeed, modifications of the zigzag model for evolution of innate immunity have been proposed that equate the RNAi response against viral pathogens to the PAMP response against nonviral pathogens [3,5]. Most or perhaps all plant viruses encode one or more proteins in their genome to suppress the effects of RNAi and protect their genomic RNA from degradation before it is encapsidated into virions [1,2]. 

A second line of defense in plants involves the capacity to recognize specific virus proteins by host resistance (R) proteins, which frequently leads to the development of a hypersensitive response (HR) [3,4,5,6,7]. HR against virus infections is typically manifested through the development of necrotic local lesions that may appear as early as two days post-inoculation (dpi) or up to 7 dpi, depending on the host/virus combination. Virus proteins recognized in this manner by host resistance proteins fit the definition of avirulence (avr) proteins in accordance with the gene-for-gene theory [3,4,5,6]. 

In this paper we have characterized the lineage of two traits associated with the coat proteins (CPs) of the tombusvirids: Silencing suppression and avirulence. Such an analysis could be valuable for understanding the interplay between plant defense elicitation and counterdefense functions associated with a pathogen effector. Members of the *Tombusviridae* are characterized by ssRNA genomes individually encapsidated into icosahedral virions. The family currently consists of 76 species, with the majority distributed within 16 genera. One genus, umbravirus, is unusual in that the viruses in this genus do not encode a CP, relying instead on the CP of their specific helper viruses [8].

Analyses of the gene order and proteins encoded by different tombusvirids indicates that one of the mechanisms that has given rise to new genera is recombination between species from different genera REFs [9,10,11,12]. Thus, genera are now defined based on the phylogenies of the complete RNA dependent RNA polymerase (RdRP) and the number of sgRNAs produced, which led to the division of genus *Carmovirus* into three new genera [9], while genus *Necrovirus* was split into *Alphanecrovirus* and *Betanecrovirus* [11] the RdRPs of alphanecroviruses and betanecroviruses lie on separate branches of RdRP phylogenetic trees, and do not reproduce the relationship of their CPs.

Several studies have shown that the CPs of this diverse family have the capacity to trigger HR in some of their hosts and/or may function as silencing suppressors. For example, the CP of the betacarmovirus turnip crinkle virus (TCV) elicits HR in *Arabidopsis thaliana* ecotype Dijon [13,14] and also functions as a silencing suppressor [15,16,17], although CP motifs conditioning silencing suppressor activity can be uncoupled from HR elicitation [18,19]. In addition to TCV, the CPs of alphacarmoviruses, gammacarmoviruses and pelarspoviruses have also been demonstrated to have silencing suppressor activity [20,21,22,23]. The CP of the tombusvirus tomato bushy stunt virus (TBSV) triggers HR in six species of *Nicotiana* belonging to section *Alatae* [24] and is not a silencing suppressor [15]. However, most tombusvirid CPs have not been evaluated for the capacity to elicit HR in *Nicotiana* species and some have not been evaluated for silencing suppression. Nonetheless, tombusvirid CPs might collectively be considered an effector, with the CP of each species comprising a structural variant within the dataset. Thus, a phylogenetic analysis of the CPs could provide insight into the evolution of a pathogen effector/avr protein.

To initially explore the avr potential of tombusvirid CPs, we focused on the capacity of the CP of tobacco necrosis virus strain D, isolate Hungary (abbreviated here as TNVD^H^), to trigger HR in *Nicotiana* species. TNVD^H^ belongs is a betanecrovirus; its genome consists of a positive-sense, single-stranded RNA of approximately 3.8 kb that is encapsidated into small icosahedral viral particles with a diameter of 28 nanometers [25,26]. The genome of TNVD^H^ encodes five proteins that are expressed through a combination of readthrough strategies and subgenomic RNAs (Figure 1A) [27]. The P22 and the readthrough product P82 are necessary for replication, the P7a and P7b proteins are necessary for movement, and the P29 protein is the coat protein, responsible for formation of the icosahedral particles [26,28]. TNVD has been transmitted experimentally to at least 88 species in 37 dicotyledonous and monocotyledonous families and it elicits HR in the majority of these species [29]. The only hosts that were initially shown to be systemically infected by TNV were *Anthriscus cerefolium* and *Trachymene cerulea* [29]. It has recently been shown that TNV also induces a systemic necrosis in *Nicotiana benthamiana* [26] and a symptomless infection in *Arabidopsis thaliana* [30]. 

The observation that TNVD triggers HR in most plant species is intriguing, because it suggests that these plants may contain receptors to recognize and defend against TNV infection. In the present study, 20 *Nicotiana* species representing the diversity of the *Nicotiana* genus were tested for their response to mechanical inoculation of TNVD^H^. We found that 19 species responded to TNVD^H^ infection with HR; in fact, TNVD^H^ was able to move systemically in only one of the species, *N. benthamiana*. To determine whether the TNVD^H^ CP was capable of triggering HR in *Nicotiana* species, we expressed the TNVD^H^ CP through agroinfiltration and found that HR is elicited in the same species of section *Alatae* that responded via HR to the TBSV CP [24]. Interestingly, a phylogenetic analysis of the tombusvirid CPs indicated that the CP of TNVD^H^ is only distantly related to the CP of TBSV [10]. Consequently, we expanded our investigation of the tombusvirid CPs to representatives of selected genera within the family, evaluating each CP for silencing suppression as well as HR elicitation in *Nicotiana*. This analysis reinforces the phylogeny of the tombusvirid CPs of the *Tombusviridae* and illustrates how structural changes that occurred in the CP effector of two of the three progenitors of the current genera generally led to either silencing suppressor activity or HR elicitation in select *Nicotiana* species.

## 2. Materials and Methods 

### 2.1. Inoculation of Nicotiana Species with TNV Virions

Seeds of different *Nicotiana* species were obtained from the U.S. Tobacco Germplasm Collection at North Carolina State University [31], as described in Angel and Schoelz [24]. To break dormancy, seeds were treated for 30 min with commercial bleach at 50% strength (2.6% *vol/vol* NaOCl). TNV- D^H^ virions were initially inoculated to *N. benthamiana* and infected tissue was frozen for further inoculations. For inoculation of test plants, plant tissues infected with TNV- D^H^ were ground with a mortar and pestle at a dilution of approximately 1:20 (*wt/vol*) with inoculation buffer (0.05 M phosphate buffer pH 7.0) and gently rubbed onto *Nicotiana* leaves dusted with 600-mesh carborundum.

### 2.2. Coat Protein Constructs

The TNV strain D^H^ clone and virions were a gift from Dr. Lorant Kiraly (Hungarian Academy of Sciences); the infectious clone was developed by Molnár and coworkers ([26]; NCBI accession number U62546). The full-length clones of CyRSV ([32]; NCBI accession number X15511) and CNV ([33]; NCBI accession number M25270) were gifts from Dr. Herman Scholthof (Texas A&M University). The full-length MCMV clone is from Dr. Kay Scheets (Oklahoma State University); its nucleotide sequence is described in Nutter et al. ([34]; NCBI accession number X14736). The full-length PMV clone was a gift from Dr. Karen Scholthof (Texas A&M University); its nucleotide sequence is described in Turina et al. ([35]; NCBI accession number U55002). The MNSV CP clone was a gift from Dr. D’Ann Rochon; its nucleotide sequence is described in Riviere and Rochon ([36]; NCBI accession number M29671.1).

The CPs of TNV D^H^, pCP∆49, pCP∆77, pCP∆146, pCP∆77KO, CyRSV, CNV, MCMV, MNSV, MNeSV, and PMV clones were all amplified by PCR, and initially cloned into pGEM-T-Easy (Promega Corp., Madison, WI). Primers used for PCR amplification were synthesized by Integrated DNA Technologies (Coralville, IA, U.S.A.) and are listed in Appendix A. PCR conditions consisted of an initial denaturation at 94 °C for 5 min, followed by 35 cycles at 95 °C for 1 min, 55 °C for 30 s, 72 °C for 1 min and a final extension at 72 °C for 5 min. The PCR product was purified by agarose gel elution using the QIAquick gel extraction kit (Qiagen Inc., Valencia, CA, U.S.A.) for cloning into pGEM-T-Easy. *Escherichia coli* colonies containing inserts were selected on Luria Bertani (LB) media containing 40 µL XGal (20 mg/mL), 10 µL IPTG (20%) and kanamycin (50 µg/mL). Candidate clones were sequenced in both orientations by the DNA Core Facility at the University of Missouri (Columbia, MO, U.S.A.). Once the fidelity of the sequence was confirmed, the insert was transferred into the *Agrobacterium* binary plasmid pKYLX7 digested with either *Xho*I or *Hind*III on the 5′ end and *Sac*I on the 3′ end [24]. Restriction enzyme sites used for each CP clone are listed in the primer sequences in Appendix A. pKYLX7 plasmids carrying the CP insert were transformed into *A. tumefaciens* strain AGL1 [37] by electroporation with a PG200 Progenetor II (Hoefer Scientific Instruments, San Francisco, CA, U.S.A.). Transformants were selected on LB medium supplemented with kanamycin (50 µg/mL) and tetracycline (12.5 µg/mL). 

The CP expression plasmids for RCNMV, TCV, PFBV, and PLPV were provided to us in *Agrobacterium* binary vectors. The RCNMV CP plasmid was a gift from Dr. Tim Sit (North Carolina State University). The RCNMV CP coding sequence (NCBI accession number J04357) was determined by Xiong et al. [38] and cloned into the Agrobacterium binary vector pPZP212 [39]. The TCV CP plasmid was a gift from Dr. Feng Qu (The Ohio State University). The cloning of the TCV CP sequence ([40]: NCBI accession number M22445) into the *Agrobacterium* binary vector PZP is described in Qu et al. [16]. The insertion of the TBSV CP construct into the Agrobacterium binary vector pKYLX7 was described previously [21]. The TBSV CP coding sequence was determined in Hearne et al. ([41]; NCBI Accession number M21958). The PFBV and PLPV CP constructs were cloned into the Agrobacterium binary vector pMOG800 under the control of the 35S promoter [21,23].

### 2.3. Phylogenetic Analysis of the Tombusvirid CPs 

The alignment was made using MUSCLE while trees were generated with the Maximum Likelihood (ML) algorithm in MEGA7 [42] using 1000 boostrap replicates. All positions with less than 50% site coverage were eliminated. That is, fewer than 50% alignment gaps, missing data, and ambiguous bases were allowed at any position.

### 2.4. Agroinfiltration Assay for HR Elicitation and Silencing Suppression

*A. tumefaciens* strains were grown in 3 Ml LB broth supplemented with kanamycin (50 μg/mL) for 24 h at 28 °C in an incubator shaker at 220 rpm. From each initial culture, 500 μL was added to flasks with 40 mL LB broth containing kanamycin (50 μg/mL), and the cultures grown for an additional 24 h. Bacteria were then sedimented by centrifugation at 14,000 g for 10 min and resuspended in 20 mL of infiltration solution (3.9 g/L MES, 20g/L Sucrose, 10 g/L Glucose, pH 5.4) supplemented with 20 μL 0.2M acetosyringone. Cells were incubated overnight at 28 °C and 220 rpm and cultures subsequently diluted to an OD_600_ 1.0 immediately before infiltration into *Nicotiana* leaves as described in Angel and Schoelz [24]. After agroinfiltration, all plants were returned to the greenhouse where they were observed on a daily basis for HR.

For the silencing suppressor assay, the virus CPs, TBSV P19, and TCV CPs were co-agroinfiltrated into *N. benthamiana* leaves at equivalent optical densities with a GFP gene that had been cloned into the binary vector pKYLX7. The GFP clone and assay are described in Angel et al. [43]. Plants were examined for GFP expression using a Blak-Ray Long Wave Ultraviolet Lamp (Upland, CA), beginning at 2 days after infiltration (dai) and extending up to 10 dai.

### 2.5. ELISA Assay for TNV CP expression 

Agroinfiltrated tissues were collected at 3 dpi and ground with mortar and pestle at a ratio of 1:3 (tissue/grinding buffer). Grinding buffer consisted of 1× phosphate buffered saline, 2% polyvinylpyrrolidone MW 40,000 g/mol, 0.2% bovine serum albumin and 0.05% Tween 20). Double antibody sandwich ELISA (DAS-ELISA) was carried out using TNV-serotype D coating and alkaline phosphatase-conjugated secondary antibodies purchased from AC diagnostics (Fayetteville, AR, U.S.A.). Colorimetric reactions with the substrate p-nitrophenyl phosphate were quantified at 405 nm using a Multiskan MCC-340 microplate reader (Thermo Fischer Scientific, Cincinnati, OH, U.S.A.). All experiments were repeated three times.

## 3. Results

### 3.1. Survey of Nicotiana Species for Resistance to TNVD^H^ Virion Inoculations

To examine the reaction of *Nicotiana* species to TNVD^H^, TNVD^H^ virions were rub-inoculated to leaves of 20 *Nicotiana* species that represent the diversity of the *Nicotiana* genus (Table 1). The same *Nicotiana* species had previously been used to characterize avr proteins present in the TBSV genome [24]. Eighteen of the twenty *Nicotiana* species responded to TNVD^H^ inoculation with a hypersensitive response, defined here as the rapid development of necrosis in the inoculated leaf. Necrotic local lesions appeared between 2 to 5 dpi and no systemic movement of the virus occurred in these plants (Table 1). Necrotic lesions varied in size, number per leaf, and timing. For example, *N. quadrivalvis* responded with large necrotic lesions that coalesced to cover nearly the entire leaf (Figure 2A), whereas *N. forgetiana* accession TW50, included in the same test, responded with fewer lesions (Figure 2B). Only one *Nicotiana* species, *N. otophora*, responded to TNVD^H^ virion inoculation with chlorotic local lesions (Figure 2C), which developed much more slowly than the necrotic responses of the other *Nicotiana* species. Interestingly, no symptoms of TNVD^H^ infection developed on upper, noninoculated leaves of *N. otophora.* It may mean that *N. otophora* is resistant to TNVD^H^ or that the plants develop a symptomless systemic infection. *N. benthamiana* was the only *Nicotiana* species scored as susceptible to TNVD^H^, as this plant species did develop systemic symptoms. *N. benthamiana* developed necrotic local lesions upon inoculation of TNVD^H^ virions (Figure 2D) and as the virus moved into upper noninoculated leaves, those leaves developed a systemic necrosis symptom (Figure 2E), as described previously by Molnár et al. [26]. Consequently, the reactions of 18 of the *Nicotiana* species were classified as an HR type resistance, one was classified as resistant with no HR, and only one species was susceptible to TNVD^H^ (Table 1). Appendix A, illustrate the diversity of responses of all 20 *Nicotiana* species to TNVD^H^ virion inoculations.

### 3.2. The TNVD^H^ Coat Protein Triggers HR in Nicotiana Species Section Alatae.

A previous study had shown that a binary plasmid *Agrobacterium tumefaciens* expressing the coat protein (CP) of Tomato bushy stunt virus elicited a rapid HR upon agroinfiltration into *Nicotiana* species belonging to section *Alatae* [24]. To investigate whether the TNVD^H^ CP was capable of triggering an HR in any *Nicotiana* species, we cloned the full-length TNVD^H^ CP coding sequence into the *Agrobacterium tumefaciens* binary vector pKYLX7 to create pTNV-CP (Figure 1). Upon agroinfiltration into leaves of each of the 20 *Nicotiana* species, pTNV-CP elicited HR in several species in section *Alatae* (Table 1), including the species *N. langsdorffii*, *N. longiflora*, *N. bonariensis*, *N. alata*, and *N. forgetiana.* Of the six species in section *Alatae*, only *N. plumbaginifolia* failed to respond to agroinfiltration of pTNV-CP with HR, even with observations up to 10 dai. By contrast, HR was initiated in *N. langsdorffii* by pTNV-CP agroinfiltration as early as 2 dai and the tissue had completely collapsed by 3 dai (Figure 3). The TNVD^H^ CP did not elicit HR in any of the other *Nicotiana* species included in our study (Table 1). The same five *Nicotiana* species that responded with HR to agroinfiltration of the TNVD^H^ coat protein also responded with HR to agroinfiltration of a plasmid expressing the TBSV CP [24]. Interestingly, the CP of TBSV also did not trigger HR in the same two accessions of *N. plumbaginifolia* (24). 

A previous analysis of the TBSV CP showed that the first 79 codons could be eliminated and the resultant CP deletion mutant was still capable of eliciting HR upon agroinfiltration into *N. langsdorffii* [21]. To determine the effect of N-terminal deletions on the capacity of pTNV-CP to trigger HR in an agroinfiltration assay, we deleted 49, 77, and 146 amino acids from the N-terminus of the TNVD^H^ coding sequence. The sizes of the deletions were determined by the presence of start codons in-frame within the TNV D^H^ coding sequence, the same strategy that was used for the choice of deletions in the TBSV CP. Both pCP∆49 and pCP∆77 triggered HR upon agroinfiltration into *N. langsdorffii* that was identical to pTNV-CP, whereas pCP∆146 did not elicit any response (Figure 3A). Furthermore, mutation of the start codon of pCP∆77 from ATG to TTG abolished HR elicitation, showing that HR was dependent on TNVD^H^ CP expression (Figure 3B). To verify that TNVD^H^ CP was expressed, total plant protein was isolated from leaves and CP was measured by ELISA. Figure 3C shows that TNVD^H^ CP epitopes were detected in leaf tissue agroinfiltrated with pCP29 and with pCP∆77. This experiment showed that the first 77 amino acids of the TNVD^H^ CP do not contribute to HR in *N. langsdorffii*, similar to an earlier finding that the first 79 amino acids of the TBSV CP do not contribute to HR in the same host [24]. 

### 3.3. Evaluation of the Coat Proteins of the Tombusviridae for Triggering HR in Members of Nicotiana Section Alatae

A phylogenetic analysis of the CPs of the type members for each genus within the tombusvirus family revealed that TBSV and TNVD^H^ CPs are distantly related (Figure 4). Since both the TBSV and TNVD^H^ CPs triggered HR in the same five *Nicotiana* species within section *Alatae*, we hypothesized that CPs of other tombusvirid genera might elicit the same response. To test this hypothesis, we compared the capacity of CP genes from representatives tombusvirid genera to trigger HR in members of section *Alatae* and also compared their ability to function as silencing suppressors in *N. benthamiana*. To assess the capacity of the CPs from tombusvirids to elicit HR in section *Alatae*, we developed or obtained CP constructs representing 10 genera within the family. We also included the CPs of two additional viruses from the tombusvirus genus: Cucumber necrosis virus (CNV) and cymbidium ringspot virus (CyRSV). 

We initially tested representatives from the tombusvirus, dianthovirus, betacarmovirus, alphanecrovirus, panicovirus, and machlomovirus genera, and the agroinfiltration tests for *N. langsdorffii* are illustrated in Figure 5. Each of the CP constructs was agroinfiltrated into leaf panels of *Nicotiana* species section *Alatae* along with pTNV-CP, to ensure that environmental conditions and leaf age were conducive for HR development, and the empty vector pKYLX7, to test for development of any nonspecific necrosis associated with infiltration of *Agrobacterium*. Each CP construct was agroinfiltrated into at least three plants and three leaves per plant, for a minimum of nine infiltrations and plants were observed up to ten days after agroinfiltration. HR was consistently elicited in the leaf panel agroinfiltrated with pTNV-CP, as well as leaf panels agroinfiltrated with constructs that expressed the CPs of RCNMV, CNV, and CyRSV. In the case of the section *Alatae* species, all responded to agroinfiltration of the same set of CPs with HR except for accessions of TW106 and 108 of *N. plumbaginifolia*. A necrotic reaction appeared at 2 to 3 dai in *N. langsdorffii*, *N. forgetiana* (TW50 and TW51) and *N. alata* (TW7 and TW8) but in the case of *N. longiflora* it usually appeared between 4–6 dai. No HR was observed in leaf panels agroinfiltrated with constructs that expressed the CPs of MCMV, TCV and PMV. The results of all of the agroinfiltration tests are summarized in Table 2.

We realized that there were significant gaps in the phylogenetic tree, and to address this issue we obtained CP clones for a gammacarmovirus, melon necrotic spot virus (MNSV), the aphacarmovirus pelargonium flower break virus (PFBV), a betacarmovirus, pelargonium line pattern virus (PLPV), and the zeavirus maize necrotic streak virus (MNeSV). In these tests, pTNV-CP was used as the positive control and pTCV-CP was used as the negative control. We found that pMNSV-CP and pMNeSV-CP constructs induced HR in *N. langsdorffii*, *N. fogetiana* and *N. alata*, but did not induce HR in *N. plumbaginifolia*. By contrast, pPLPV-CP and pPFBV-CP did not induce HR in any of the *Nicotiana* species in section *Alatae* (results illustrated for *N. langsdorffii*, Suppl. Figure 5, Table 2). 

An inspection of the phylogenetic tree constructed for tombusvirid CPs shows that Clades 2 and 4 have retained the capacity to trigger HR in members of *Nicotiana* section *Alatae* (Figure 4). HR elicitation was confirmed for two CPs within Clade 2: The betanecrovirus TNV D and the zeavirus MNeSV. Furthermore, HR elicitation was also confirmed within Clade 4 for five CP sequences within the genera: three tombusviruses (TBSV, CNV, CyRSV), the dianthovirus RCNMV, and the gammacarmovirus MNSV. 

### 3.4. Evaluation of the CPs of Selected Tombusvirids for a Functional Silencing Suppressor in N. benthamiana

CPs of several of the tombusvirids have been shown to function as a silencing suppressor: betacarmoviruses TCV and Hibiscus chlorotic ringspot virus (HCRSV) [16,17,22], alphacarmovirus PFBV [21], gammacarmovirus MNSV [20], and pelarspovirus PLPV [23]. In each case, GFP expression has been enhanced and extended when a binary plasmid designed to express GFP is co-agroinfiltrated into *N. benthamiana* leaves with plasmids containing one of these CPs. By contrast, the CP of MCMV does not act to enhance and extend the expression of GFP in the standard silencing suppressor assay that works for HCRSV, PFBV, MNSV, TCV and PLPV [44]. 

To investigate whether the TNV CP had the capacity for silencing suppression, we co-agroinfiltrated pTNV-CP with p35S-GFP into *N. benthamiana* leaves and evaluated GFP expression over a period of 8–10 days. GFP expression in leaf sections agroinfiltrated with p35S-GFP alone or co-agroinfiltrated with pTNV-CP peaked at 3–4 dai and by 8 dai, GFP expression was extinguished (Figure 6). By contrast, GFP expression remained strong in leaf sections co-agroinfiltrated with p35S-GFP and plasmids containing TCV CP or TBSV P19. This test showed that TNV D^H^ CP does not function as a silencing suppressor analogous to TCV CP or TBSV P19. To our knowledge, the CPs of several other tombusviruses have not been formally evaluated for silencing suppression. We found that none of the CPs of CNV, MCMV, PMV, CyRSV or MNeSV displayed silencing suppressor activity (Figure 6, Table 2). Of the tombusvirid CPs previously found to be silencing suppressors, we confirmed this activity for the CPs of PLPV, PFBV, and MNSV, whereas the CPs of TBSV and RCNMV were confirmed to have no silencing suppressor function (Appendix A, Table 2). In tracing the lineage of silencing suppressor function for the tombusvirid CPs, the trait appears to be confined to Clade 3, with the one exception of the gammacarmovirus CP of MNSV, which is located in Clade 4 (Figure 4). 

## 4. Discussion

Agroinfiltration is a powerful tool for the discovery and initial characterization of pathogen proteins capable of triggering HR in plant hosts. This tool has been validated for several virus avr proteins including the cauliflower mosaic virus P6 protein [45] and the helicase domain of the tobacco mosaic virus (TMV) replicase [46,47], the NSm gene of tomato spotted wilt virus (TSWV) [48] as well as the P19 and P22 proteins of TBSV [40]. For each of these virus *avr* genes, the avr trait was first identified through techniques involving either gene swaps between virus strains [49,50], the insertion of the *avr* gene into a virus vector [51,52], or direct mutagenesis of the *avr* gene within an infectious clone of the virus [53]. The capacity for HR was then confirmed through agroinfiltration of the viral gene, separate from the virus genome. Agroinfiltration has also been used as the initial technique for characterization of virus *avr* genes. For example, it has been used to show that the NSs protein of TSWV triggers HR in *Capsicum annuum* species resistant to the virus infection [54]. Furthermore, the CP gene of Potato virus X (PVX) was shown to induce HR upon its co-agroinfiltration with its *R* gene counterpart into *N. benthamiana* [55]. Recently, Vleeshouwers and coworkers [56] utilized agroinfiltration to initially characterize the capacity of 54 effectors of *Phytophthora infestans* to trigger HR in wild *Solanum* species, illustrating how this technique could accelerate discovery and functional analysis of pathogen *avr* genes. 

In a previous paper, Angel and Schoelz [24] showed that the TBSV CP triggered HR in several members of *Nicotiana* section *Alatae*, including *N. alatae*, *N. langsdorffii*, *N. forgetiana*, *N. bonariensis*, and *N. longiflora*. In the present paper we have now extended this analysis to show that the same *Nicotiana* species in section *Alatae* that recognized the TBSV CP also recognize the CPs of several other tombusvirids. This analysis has implications for the lineage of the avr motif (or motifs) associated with the tombusvirid CPs as well as the lineage for the silencing suppressor function associated with the CP.

### 4.1. Tracing the Lineage of HR Induction and Silencing Suppression Associated with the CPs of the Tombusviridae

In this paper we considered the CPs of individual tombusvirid species to be structural variants of a single effector. Consequently, a phylogenetic tree of the CPs can be a valuable source of information on the traits associated with the CP, just as phylogenetic trees of host plants can be informative about the inheritance of resistance. The two traits we evaluated in this paper were the capacity to trigger HR in *Nicotiana* species of section *Alatae* and silencing suppression. Silencing suppressor activity associated with the tombusvirid CP has been characterized in several papers [16,17,20,21,22,23]. Our goal in the present paper was to characterize the status of silencing suppressor activity in virus species for which there are no published records. Furthermore, we considered it valuable to test silencing suppressor activity and HR elicitation of CP constructs under a uniform set of conditions. 

We found that elicitation of HR in *Nicotiana* could be traced to CPs in two clades: Clades 2 and 4 (Figure 4). In Clade 2, elicitation of HR was confirmed in one betanecrovirus and the monotypic zeavirus (Figure 4); it remains to be seen whether the CPs of the alphanecroviruses also have the capacity to trigger HR in *Nicotiana*. In Clade 4, HR elicitation in *Nicotiana* was confirmed by one or more tombusvirus, gammacarmovirus, and dianthovirus, but must still be investigated in aureusviruses, and the single gallantivirus, and macanavirus (Figure 4). Further work is necessary to determine whether an amino acid motif common to the CPs in Clades 2 and 4 is recognized by a single R protein in *Nicotiana*, or whether recognition is mediated by motifs unique to each of the clades. 

An intriguing result is that viruses in Clade 1, which are the only tombusvirids that produce small CPs lacking a protruding domain, had neither HR elicitation nor silencing suppressor function, and hosts for MCMV and panicoviruses are restricted to the family *Poaceae*. MNeSV is likewise restricted to hosts in the family *Poaceae*, but nt sequence analysis indicated that the homology to tombusviruses surrounding the necrovirus-like CP ORF had borders that precisely retain two tombusviral regulatory sequences [12]. Thus, it is apparent that MNeSV has not lost the betanecrovirus CP characteristics that induced necrosis. The host range of the macanavirus furcrea necrotic streak virus is restricted to members of the monocot family *Asparagaceae*, [57] which may limit the ability of its CP, which contains a protruding domain, to cause any effect in dicot species.

In tracing the lineage of silencing suppression within the CP, we found that the silencing suppressor function was largely confined to Clade 3 (Figure 4). A comparison of the CPs in Clades 3 and 4 show that the silencing suppression trait in Clade 3 can be genetically separated from the HR determinant in Clade 4. In fact, the phylogenetic tree for the CP suggests that the separation of these traits occurred in a progenitor that led to the occurrence of Clades 3 and 4 (Figure 4, red arrow). It is significant to note that silencing suppressor function has been attributed to proteins other than the CP in many of the virus species in Clade 4, such as the strong silencer P19 in the tombusvirus genus [2] and both the RCNMV replication complex and MP [58,59]. Consequently, the CPs of the viruses in Clade 4 might not have a need to function as silencing suppressors.

The one intriguing exception to the separation of silencing suppressor and HR determinant is found in the CP of MNSV, as it carries both traits. Interestingly, phylogenetic analyses of the RdRP and movement proteins of tombusvirids showed that these proteins are more closely aligned with the other members of the gammacarmovirus genus in Clade 3 than with any member of Clade 4 [10], suggesting a recombination event occurred between MNSV and some member of Clade 4 to orient the MNSV CP in Clade 4. Indeed, interfamilial recombination has already been documented between the 3′ untranslated region of MNSV and the polerovirus cucurbit aphid-borne yellows virus, a luteovirid [60], so it is possible that other recombination events might have placed the MNSV CP ORF in Clade 4 and the balance of the virus in Clade 3. 

The MNSV CP demonstrates that although separate, the motifs for HR elicitation in *Nicotiana* and silencing suppression can coexist in the same protein sequence. This conclusion is similar to what has been found with the CP of TCV, which also carries motifs for silencing suppression and HR elicitation in *A. thaliana* ecotype Dijon [13,14,15,16,17]. Choi and coworkers [18] also concluded that HR elicitation and silencing suppression were separate traits. Interestingly, the motif in the TCV CP that elicits HR in Dijon is likely located in a different part of the CP than the motif in the CPs that triggers HR in *Nicotiana*. In the case of the TCV CP, the amino acids responsible for breaking resistance in Dijon have been localized to the N-terminus of the TCV CP [14,61]. By contrast, deletion analyses of the CPs of TBSV and TNVD^H^ have shown that the capacity to trigger HR in *Nicotiana* section *Alatae* is retained in CPs in which the first 77 amino acids can be deleted.

The interplay between resistance and susceptibility is frequently portrayed as an arms race between the pathogen and host in which any advantage gained by one side is countered by modifications on the other to nullify that advantage [3,5,62]. Similarly, it has also been suggested that viral silencing suppressors may be preferentially targeted by the host, perhaps to protect the integrity of the host silencing apparatus [3,52,54,62,63]. In this scenario, host R proteins have arisen to counter the silencing suppression function of the virus protein. However, the phylogenetic analysis in Figure 4 is striking because it indicates that there may not be any obvious relationship between the development of the CP as a silencing suppressor and its selection as a target for HR elicitation by the host. With the exception of MNSV, the CPs in Clade 4 that are the target for HR elicitation by members of *Nicotiana* Section *Alatae* have all lost the capacity of silencing suppression, whereas an R protein has not yet been identified that can uniformly recognize the CPs that retain silencing suppressor function. An alternative hypothesis would be that the silencing suppression and avr traits may be associated with structural requirements of the virion, traits that became set in the CP progenitors that resulted in the separation of Clades 3 and 4. 

The three-dimensional structure of the CPs of several tombusvirids have been determined (reviewed in [64]), and coupled with the wealth of amino acid sequence information on tombusirid CPs, it would be valuable now to further examine the structural basis for silencing suppression and for elicitation of HR. In fact, amino acid sequences associated with the silencing suppressor function of the TCV CP were characterized in Cao et al. [19]; in particular, mutation of two basic amino acid residues (R130 and R137) appeared to have a significant effect on silencing activity. With the increased number and diversity of CP sequences now available, it would be interesting to continue with this analysis. It is possible that both silencing suppression and HR elicitation may be associated as much with the secondary and tertiary structure of the CP rather than with the primary amino acid sequences. A similar analysis of the CP effector of TMV was completed several years ago, showing that the three-dimensional structure of the coat protein is critical for recognition by the *N’* gene. The recognition site consists of a central hydrophobic core surrounded by polar and charged amino acid resides [65]. In addition, the formation of coat protein dimers, trimers, and tetramers may also influence recognition [66]. It would be interesting to know if the sequence in the CPs of the members of Clades 2 and 4 recognized by at least one putative R protein in *Nicotiana* also is as complex as the TMV CP elicitor.

### 4.2. Tracing the Lineage of Resistance Genes to the Tombusviridae in the Genus Nicotiana

Many *Nicotiana* species respond to TBSV and TNV virion inoculation with HR, and putative R genes within these species target at least three different TBSV proteins (24). The P19 protein triggers HR in *N. tabacum*, *N. sylvestris*, and *N. bonariensis* [24,51]. Similarly, the P22 gene triggers HR in *N. glutinosa* and *N. forgetiana* TW50 [24,51]. Since most tombusvirids do not carry genes comparable to P19 and P22, these R proteins would appear to be targeting primarily members of the genus tombusviruses and aureusviruses. However, a third type of R gene has evolved to target a broader range of the tombusvirids, as our evidence indicates that several species of *Nicotiana* section *Alatae* can recognize the CPs of tombusviruses [24], dianthovirus, gammacarmovirus, betanecrovirus and zeavirus. 

These results illustrate the potential for characterizing resistance genes in *Nicotiana* species towards different viruses and strains of the same virus. For instance, Doroszewska and Depta [67] inoculated virions of six isolates of Potato virus Y (PVY) from three groups, PVY^NW^, PVY^NZ^ and PVY^NTN^, on leaves of 96 accessions of 68 *Nicotiana* species, including autotetraploid forms and botanical varieties. Five accessions belonging to *N. africana*, *N. glauca*, *N. raimondii*, *N. knightiana* and *N. benavidesii* were fully resistant to the six PVY isolates, but several other accessions and species were resistant to one or more PVY isolates, varying in their response. Similarly, Laskowska et al [68] tested 94 accessions of *Nicotiana* species accessions against TSWV and found that members of the *Alatae* section were mostly resistant, including five accessions in *N. alata*, *N. forgetiana*, three accessions of *Nicotiana* x *sanderae*, and also two *N. tabacum* cultivars showed HR local or systemic related resistance to TSWV.

Interestingly, a recent study has shown that several members of *Nicotiana* section *Alatae* also carry a functional orthologue of the *N’* gene, which recognizes the CP of TMV [69]. Members of section *Alatae* that carry a functional orthologue include *N. alatae* (PI42334), *N. forgetiana*, and *N. langsdorffii*. It is intriguing that these *Nicotiana* species are able to respond with HR to inoculation of viruses from such genera as tobamovirus, tombusvirus, and betanecrovirus, and that these species respond to an even larger selection of CPs with HR upon agroinfiltration. It would be interesting to know if there is a common R protein in these *Nicotiana* species responsible for recognition these viruses. It may be that an *N* or *N’* homologue might condition resistance in *Nicotiana* to tombusvirids. Yuan and coworkers [69] reported that *Nicotiana* species carried from one to up to 22 distinct *N* homologues, whereas Balaji et al. [70] found evidence for approximately 14 *N* homologues in *N. clevelandii* and *N. glutinosa*, in addition to the N gene. Significantly, silencing of the *N* gene present in *N. edwardsonii* suppressed HR induced by inoculation of TBSV and CymRSV, as well as TMV, suggesting that a *Tombusvirus* R gene shares some homology with the *N* gene. 

## Figures and Tables

**Figure 1 viruses-11-00588-f001:**
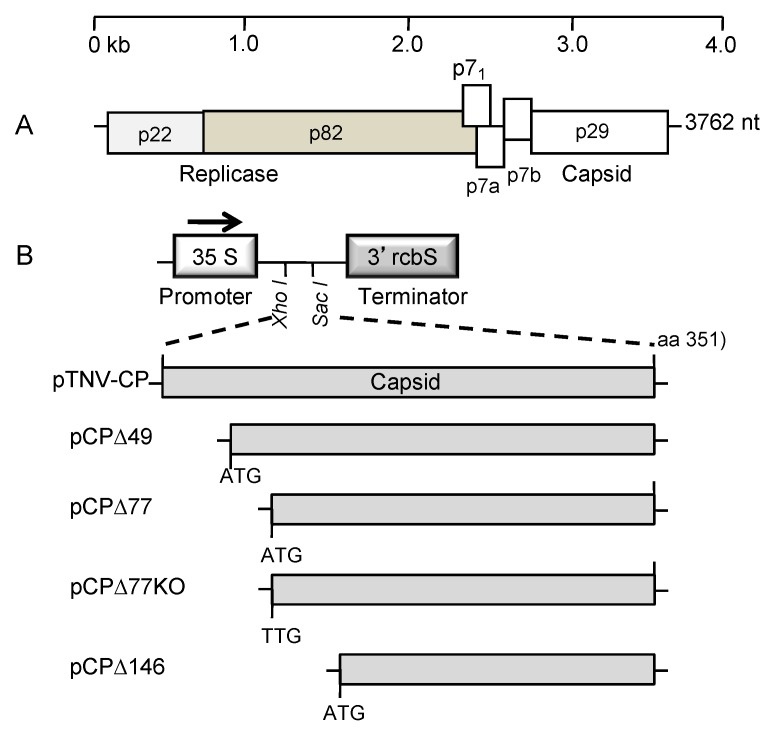
Genome organization of TNVD^H^, and TNV CP constructs used for transient expression in *Nicotiana* species. (**A)** TNV-D^H^ genome structure. The six open reading frames are illustrated by boxes. The p82 protein is highlighted in a different color from the p22 protein to emphasize that it is a readthrough product of the p22 protein. (**B)** Structure of inserts into the *Agrobacterium* binary vector pKYLX7. The arrow indicates the direction of transcription of the 35S promoter. All TNV CP constructs were generated by PCR and delimited by *Xho*I and *Sac*I sites for cloning into pKYLX7. The clones pCP∆49, pCP∆77, and pCP∆146 utilized start codons present in-frame within the TNVD^H^ CP coding sequence, and represent deletion of the first 49, 77 and 146 codons of the CP sequence, respectively. The clone pCP∆77KO, the start codon has been changed to TTG.

**Figure 2 viruses-11-00588-f002:**
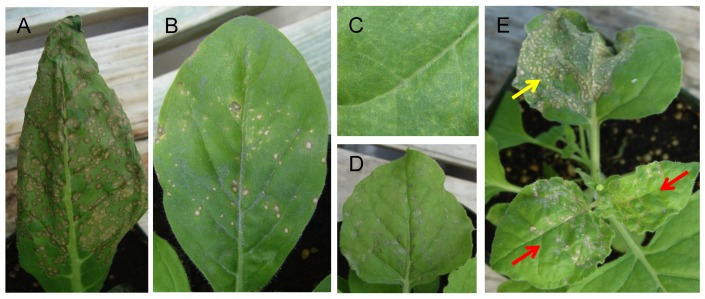
Response of *Nicotiana* species to TNVD^H^ virion inoculation. (**A**) *N. quadrivalvis* at 3 dpi. (**B**) *N. forgetiana* TW50 at 5 dpi. (**C**) *N. otophora* at 3 dpi. (**D**) *N. benthamiana* at 5 dpi. (**E**) *N. benthamiana* at 9 dpi. The red arrows indicate *N. benthamiana* leaves exhibiting systemic symptoms, whereas the yellow arrow indicates lesions in an inoculated leaf.

**Figure 3 viruses-11-00588-f003:**
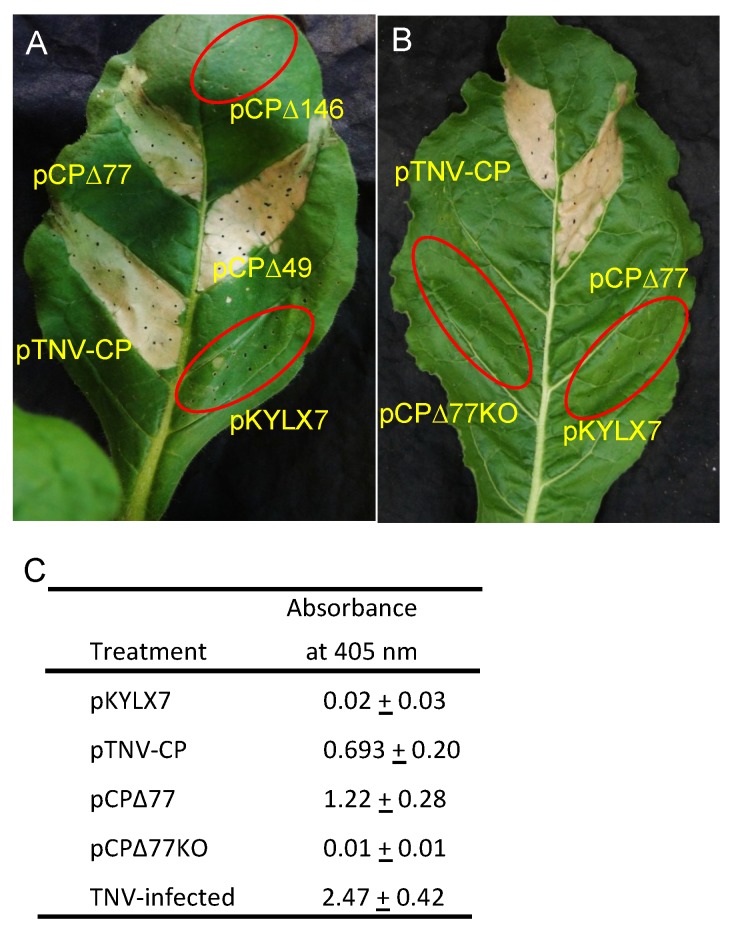
Agroinfiltration of pTNV-CP, N-terminal TNV CP deletion mutants, and empty vector pKYLX7 into *N. langsdorffii* at 3 dai. A. Analysis of TNV-CP and three deletion mutants. B. Analysis of TNV-CP, pCP∆77 and pCP∆77KO. C. ELISA values assessed at 405 nm for expression of TNV CP constructs. Leaf panels infiltrated with constructs that do not react with HR are circled in red.

**Figure 4 viruses-11-00588-f004:**
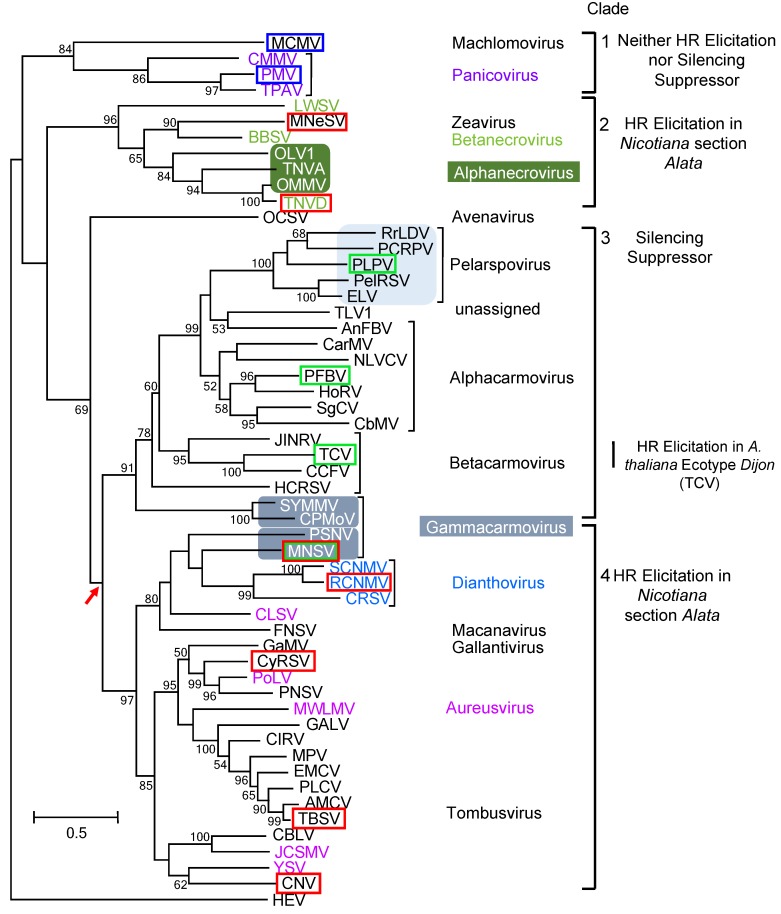
Phylogenetic (distance) analysis of 54 CPs of CP-encoding tombusvirids. ML trees were generated as described in materials and methods. There were 356 positions in the final dataset. Hepatitis E virus (HEV) CP (AAA03191.1) was used as the outgroup. Brackets and/or colored text and/or colored boxes mark monophyletic RdRP lineages. Virus CPs enclosed in a red box were identified as avirulence determinants for the *Nicotiana* section *Alatae*. Virus CPs enclosed in a green box function as silencing suppressors in *N. benthamiana*. Virus CPs in blue boxes neither triggered HR in members of section *Alatae* nor acted as silencing suppressors. The arrow indicates a point of divergence between CPs that elicit HR in *Nicotiana* section *Alata* and CPs that function as silencing suppressors.

**Figure 5 viruses-11-00588-f005:**
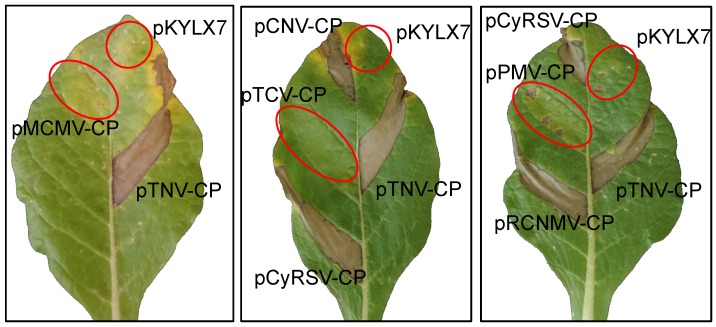
Evaluation of selected CPs of the *Tombusviridae* for elicitation of necrosis in *N. langsdorffii*. pTNV-CP and empty vector pKYLX7 were agroinfiltrated into every leaf as positive and negative controls, respectively. Red circles illustrate zones of infiltration for constructs that did not elicit necrosis. Photos were taken at 7 dai.

**Figure 6 viruses-11-00588-f006:**
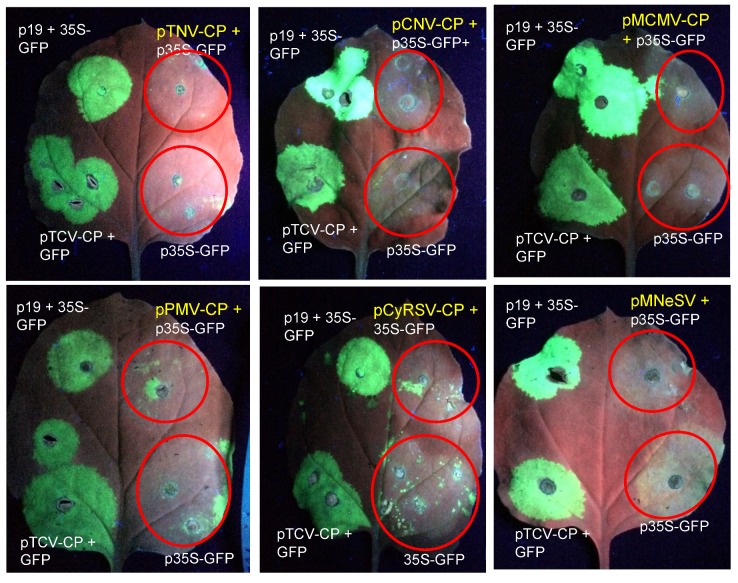
Agroinfiltration of p35S-GFP alone or co-agroinfiltration with virus CP clones or TBSV p19. The clones pTCV-CP and TBSV p19 were included in each leaf as positive controls for silencing suppression, whereas p35S-GFP was included in each leaf to illustrate the induction of the host silencing response. The zone of infiltration is highlighted in tissues that did not exhibit silencing suppression. Photos were taken under UV illumination between 8–10 dai.

**Table 1 viruses-11-00588-t001:** Response of 20 *Nicotiana* species to inoculation with TNV virions and agroinfiltration of the pTNV-CP.

*Nicotiana* spp.	Section	TNV Virion Inoc.a	Agroinfiltration of pTNV-CP
*N. langsdorffii*	*Alatae*	HR ^b^	N ^c^
*N. longiflora*	*Alatae*	HR	N
*N. bonariensis*	*Alatae*	HR	N
*N. alata*	*Alatae*	HR	N
*N. forgetiana*	*Alatae*	HR	N
*N. plumbaginifolia*	*Alatae*	HR	no rxn ^d^
*N. quadrivalvis*	*Polydicliae*	HR	no rxn
*N. clevelandii*	*Polydicliae*	HR	no rxn
*N. edwardsonii ^d^*	*Undulatae/Polydicliae*	HR	no rxn
*N. glutinosa*	*Undulatae*	HR	no rxn
*N. arentsii*	*Undulatae*	HR	no rxn
*N. undulata*	*Undulatae*	HR	no rxn
*N. tabacum*	*Nicotiana*	HR	no rxn
*N. sylvestris*	*Sylvestres*	HR	no rxn
*N. otophora*	*Tomentosae*	CLL ^e^	no rxn
*N. tomentosiformis*	*Tomentosae*	HR	no rxn
*N. repanda*	*Repandae*	HR	no rxn
*N. glauca*	*Noctiflorae*	HR	no rxn
*N. rustica*	*Rusticae*	HR	no rxn
*N. benthamiana*	*Suaveolentes*	Susc ^f^	no rxn

^a^ Virions were present in sap from *N. benthamiana* leaves infected with TNV. ^b^ HR, necrotic local lesions, no development of systemic symptoms. ^c^ N, rapid necrosis within the zone of infiltration. ^d^ no rxn, no visible reaction within the zone of infiltration. ^e^ CLL, chlorotic local lesions, no development of systemic symptoms. ^f^ Susceptible – symptoms develop in upper, non-inoculated leaves.

**Table 2 viruses-11-00588-t002:** Reaction of *Nicotiana* species in section *Alatae* to agroinfiltration of CPs of the *Tombusviridae* and assessment of silencing suppressor function of CPs in *N. benthamiana.*

Virus CP	*N. langsdorffii*	*N. longiflora*	*N. alata* tw7	*N. forgetiana* tw50	*N. plumbaginifolia* tw106	Silencing Suppressor Activity ^a^
tombusvirus						
pTBSV	N ^b^	N	N	N	no rxn ^c^	No
pCyRSV	N	N	N	N	no rxn	No
pCNV	N	N	N	N	no rxn	No
gammacarmovirus						
pMNSV	N	N	N	N	no rxn	Yes
dianthovirus						
pRCNMV	N	N	N	N	no rxn	No
betacarmovirus						
pTCV	no rxn	no rxn	no rxn	no rxn	no rxn	Yes
pelarspovirus						
pPLPV	no rxn	no rxn	no rxn	no rxn	no rxn	Yes
alphacarmovirus						
pPFBV	no rxn	no rxn	no rxn	no rxn	no rxn	Yes
zeavirus						
pMNeSV	N	N	N	N	no rxn	No
alphanecrovirus						
pTNV	N	N	N	N	no rxn	No
panicovirus						
pPMV	no rxn	no rxn	no rxn	no rxn	no rxn	No
machlomovirus						
pMCMV	no rxn	no rxn	no rxn	no rxn	no rxn	No

^a^ Silencing suppressor activity assessed by co-agroinfiltration of target CP with 35SGFP in N. benthamiana. ^b^ N, rapid necrosis within the zone of infiltration. ^c^ no rxn, no visible reaction within the zone of infiltration.

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
