# Peer review of "Tracing the Lineage of Two Traits Associated with the Coat Protein of the Tombusviridae: Silencing Suppression and HR Elicitation in Nicotiana Species"

_viruses, 2019, doi:10.3390/v11070588_

Reviewer 1 Report

This is a valuable work focused on the identification/confirmation of two traits associated to CPs of members of family Tombusviridae (RNA silencing suppression and avirulence) and their distribution through a phylogenetic tree. To this aim,authors consider all CPs as an effector and the CP of each species a structural variant within the data set. Experiments are well conducted and results may lay the foundations for future research on the evolution of the two traits and on the identification of motifs associated to one or the other.

Some points should be addressed:

- Lines 181-182. "...N. otophora also was resistant...as no symptoms developed on upper, non-inoculated leaves...". No symptoms does not necessarily means lack of infection; the presence/absence of the virus should be confirmed by any molecular method.

- Number of genera in Tombusviridae is confusing along the text (including the abstract). Please clarify according to ICTV.

- As authors indicate, MNSV CP makes a difference with regard other CPs as suggests that HR elicitation in Nicotiana and silencing suppression may coexist in the same protein. Nevertheless, in reference 17, MNSV CP is defined as a weak silencing suppressor which contrasts with results obtained by authors (supplemental Fig. 6). Any explanation for this?. This could be a non-negligible point as a weak suppressor function could mean the loss of a function in favor of the other or the recent acquisition of a function that needs to be improved.

- Line 87. Nicotiana benthamina in italics.

Author Response

We wish to thank editor and the reviewers for the time spent reviewing our manuscript.  I have reproduced the reviewers’ comments below, with a detailed response to each of their comments. The reviewers’ comments were thought-provoking, and we hope that our responses will have answered their concerns.  In the manuscript, the changes that were made in response to the reviewers’ comments are highlighted in yellow. 

Both reviewers had questions about the family tombusviridae.  For this reason, we invited Kay Scheets to become a co-author of the paper and she is listed now on the title page.  We had been working with Kay for a number of years, initially on MNeSV and then on MCMV.  After extensive discussions about CP phylogeny, Kay generated a new phylogenetic tree for Fig. 4 and added the details for its creation in a new section in the Materials and Methods.  She was instrumental in addressing Reviewer #2’s questions about the tree and also in helping us with the taxonomy of the tombusvirdae.  She suggested several new references and these are now highlighted in the reference section.  Kay also is an expert on the nomenclature of the tombusviridae, so extensive modifications were made regarding the italicization of names. These changes are not highlighted. Kay has made essential contributions to this paper and her inclusion on this paper is justified. 

Reviewer #1

This is a valuable work focused on the identification/confirmation of two traits associated to CPs of members of family Tombusviridae (RNA silencing suppression and avirulence) and their distribution through a phylogenetic tree. To this aim,authors consider all CPs as an effector and the CP of each species a structural variant within the data set. Experiments are well conducted and results may lay the foundations for future research on the evolution of the two traits and on the identification of motifs associated to one or the other.

Some points should be addressed:

- Lines 181-182. "...N. otophora also was resistant...as no symptoms developed on upper, non-inoculated leaves...". No symptoms does not necessarily means lack of infection; the presence/absence of the virus should be confirmed by any molecular method.

Response:  We agree with the reviewer.  We now state that, “Interestingly, no symptoms of TNV-DH infection developed on upper, non-inoculated leaves of N. otophora. It may mean that N. otophora is resistant to TNV-DH or that the plants develop a symptomless systemic infection.”  Nonetheless, our agroinfiltration screen for avirulence proteins only works for viruses that trigger HR, so although we cannot state that N. otophora is resistant, such knowledge does not further the narrative of the paper.  Instead we choose to be more careful in how we describe the reaction of N. otophora to TNV-DH infection.  

- Number of genera in Tombusviridae is confusing along the text (including the abstract). Please clarify according to ICTV.

Response: we agree with the reviewer. Our narrative is confusing.  To simply this, we eliminated the reference to number of genera in the abstract, as this sentence was not necessary.  We also clarified the number of genera mentioned in the Introduction.

- As authors indicate, MNSV CP makes a difference with regard other CPs as suggests that HR elicitation in Nicotiana and silencing suppression may coexist in the same protein. Nevertheless, in reference 17 (now ref 20), MNSV CP is defined as a weak silencing suppressor which contrasts with results obtained by authors (supplemental Fig. 6). Any explanation for this?. This could be a non-negligible point as a weak suppressor function could mean the loss of a function in favor of the other or the recent acquisition of a function that needs to be improved.

Response: Genoves et al. (2014) used a slightly different assay than us. In their assay, the GFP reporter and MNSV CP silencing suppressor were infiltrated into N. benthamiana line 16C (transgenic for GFP). They found that their construct failed to suppress silencing between 7 and 11 dai, whereas HC-Pro was actively suppressing silencing of GFP at 11 dai.  For this reason, they classified the MNSV CP as a weak silencing suppressor.  In our assay, the photo of the MNSV CP silencing suppressor was taken at 8 dai, which was enough time to confirm silencing suppressor function but not enough time to address the issues raised by the reviewer.  Our sense is that the silencing suppressor function of our MNSV CP construct would have been extended beyond 11 dpi, but it did not occur to us to continue with the experiment.  Unfortunately, we do not have any N. benthamiana in the greenhouse now to test this hypothesis and it would take approximately two months to generate these plants.  Although the reviewer raises an interesting point, we could not find a place to introduce this into the Discussion without interrupting the flow of the narrative.

- Line 87. Nicotiana benthamina in italics.

Response: N. benthamiana has been changed to italics. Thank you for catching this.

Reviewer 2 Report

This paper investigates CP function of Tombusviridae in light of silencing suppression and HR elicitation, and discuss their evolutional lineage in Nicotiana species in section Alatae. The authors phylogenetically classified CP into four clades and showed that 1) silencing suppression with no HR elicitation in clade3, 2) HR elicitation with no silencing suppression in clades2,4, 3) no silencing suppression and no HR elicitation in clade1 based on the results from transient expression of each CP separated from the context of virus infection. This work is interesting and appears to have been carried out soundly, but some experiments are required to draw the authors’ conclusion.

1.In the assay for HR elicitation using agroinfiltration(Table1,2; Figure 3,5; Supplementary Figure5), accumulation of CP in protein level should be confirmed to conclude that it does NOT induce cell death because agroinfiltration does not always effectively express test gene in any plants species and no visible phenotype might be due to low accumulation of the tested CP. 

2.CP accumulation should be investigated in the assay for RNA silencing suppression(Figure6; Supplementary Figure6). Furthermore, RNA silencing is triggered against mRNA in sense mRNA-mediated PTGS (GFP over-expression in this study), therefore confirmation of GFP mRNA levels by northern blot or real-time PCR is required.

Minor points

1.line154: please describe agrobacterium strain. AGL1?

2.Table1: According to Materials and Methods, this experiment was conducted using sap from N. benthamiana infected with TNV, not using virion itself.

3.Figure 4: TNVD indicates TNV?

4.Figure 4: Phylogenetic tree in Scheets et al., 2014 is different from that in Figure 4. Did the authors re-analyze phylogenetic relationship of the CPs?  If so, please describe the method to reconstruct phylogenetic tree in Materials and Methods.

5.Table2: Did the authors check the reaction of Nicotiana species in section Alatae against infection of viruses that used in Table2?

6.lines, 273-275: The sentence “Most notably, ...” should be moved to discussion section because readers cannot understand what it means without prior information.

7.lines, 304-319: this section can be omitted or shortened because the strategies and methods to validate elicitor molecules for HR are well-known.

8.lines 374-403/406-414: evolutional and molecular insights on arms race between Nicotiana species and Tombusviridae are a little difficult to follow. I recommend re-organization. 

Author Response

We wish to thank editor and the reviewers for the time spent reviewing our manuscript.  I have reproduced the reviewers’ comments below, with a detailed response to each of their comments. The reviewers’ comments were thought-provoking, and we hope that our responses will have answered their concerns.  In the manuscript, the changes that were made in response to the reviewers’ comments are highlighted in yellow. 

Both reviewers had questions about the family tombusviridae.  For this reason, we invited Kay Scheets to become a co-author of the paper and she is listed now on the title page.  We had been working with Kay for a number of years, initially on MNeSV and then on MCMV.  After extensive discussions about CP phylogeny, Kay generated a new phylogenetic tree for Fig. 4 and added the details for its creation in a new section in the Materials and Methods.  She was instrumental in addressing Reviewer #2’s questions about the tree and also in helping us with the taxonomy of the tombusvirdae.  She suggested several new references and these are now highlighted in the reference section.  Kay also is an expert on the nomenclature of the tombusviridae, so extensive modifications were made regarding the italicization of names. These changes are not highlighted. Kay has made essential contributions to this paper and her inclusion on this paper is justified. 

Reviewer #2

This paper investigates CP function of Tombusviridae in light of silencing suppression and HR elicitation, and discuss their evolutional lineage in Nicotiana species in section Alatae. The authors phylogenetically classified CP into four clades and showed that 1) silencing suppression with no HR elicitation in clade3, 2) HR elicitation with no silencing suppression in clades2,4, 3) no silencing suppression and no HR elicitation in clade1 based on the results from transient expression of each CP separated from the context of virus infection. This work is interesting and appears to have been carried out soundly, but some experiments are required to draw the authors’ conclusion.

1.In the assay for HR elicitation using agroinfiltrationTable1,2; Figure 3,5; Supplementary Figure5, accumulation of CP in protein level should be confirmed to conclude that it does NOT induce cell death because agroinfiltration does not always effectively express test gene in any plants species and no visible phenotype might be due to low accumulation of the tested CP. 

2.CP accumulation should be investigated in the assay for RNA silencing suppressionFigure6; Supplementary Figure6. Furthermore, RNA silencing is triggered against mRNA in sense mRNA-mediated PTGS (GFP over-expression in this study), therefore confirmation of GFP mRNA levels by northern blot or real-time PCR is required.

Response to points 1 and 2: The reviewer raises two valid issues. The first issue raised by the Reviewer is that it is necessary to provide evidence of expression of each of the CP constructs. This is as much necessary for silencing suppression as HR elicitation.

We have added data on the expression of the TNV CP, as well as on expression of a CP knockout construct. This information has been incorporated into the narrative and into Fig 1 and Fig. 3.    

However, we also argue here that the bioassays we use for HR elicitation and silencing suppressor function are sufficient to prove expression of the CP for our paper.  We believe that the reviewer has not considered the characterizations of CP constructs in prior publications or precendents in the literature for how pathogen elicitors are evaluated in other systems.   

Several of the CPs of the viruses we used in this paper triggered HR in section Alatae upon agroinfiltration.  These viral CPs include TBSV, TNV-D, CNV, CyRSV, RCNMV, MNeSV, and MNSV.  We characterized the expression of the TBSV CP in Angel and Schoelz (2013) and have added information for TNV CP in the present manuscript. We believe that similar HR results for CNV, CyRSV, RCNMV, MNeSV and MNSV are confirmation of expression of those viruses.

The authors specifically raise the issue that a negative result might be due to low accumulation of the CP.  The results obtained with MSNV are especially important, because this construct was positive for both HR elicitation and silencing suppression. Although the silencing suppressor function of MNSV appears comparable to TCV, PRBV, and PLPV, MNSV CP could also trigger HR, in contrast to the other three.  This result argues that there is a structural basis for the differences in HR elicitation rather than modulations in expression. MNSV is also an important control in evaluating the lack of silencing suppression of all of the viruses in Clade 3, as it triggers a robust HR, similar to other members of Clade 3.

Finally, we also believe our approach is sound, based on the practices used in previous papeers.  We cite three precedents in the literature in which the expression of constructs designed for agroinfiltration were not confirmed through analysis of proteins. 

1.     Sophien Kamoun’s lab (ref. 56) published an analysis on 54 effectors of P. infestans in 2008. Each of the effectors was expressed in a PVX vector, and there is no mention that the expression levels were evaluations.  This approach illustrates how transient expression systems could accelerate discovery and analysis of pathogen avr genes. 

2.     Martinez-Turiño and Hernandez (2009) (ref 21) analyzed the genome of PFBV for silencing suppressors through agroinfiltration of individual virus genes expressed from the 35S promoter to find that the PFBV CP is a silencing suppressor.  No protein expression data is presented.

3.     Genovés et al. (2006) (ref. 20) agroinfiltrated all of five of the ORFs of MNSV into N. benthamiana to show that its CP is a silencing suppressor.  No protein data was shown for any construct.

There are probably other examples in our reference list, and if necessary, we would be willing to check further. We agree that protein expression data would add extra proof of expression, but there are ample precedents in the literature for our approach.  

The second issue raised by the reviewer is that it is necessary to provide northern blots or real-time PCR to confirm GFP mRNA levels.  We believe that our bioassays are sufficient.  All of the CP silencing suppressors included in this study had already been characterized and the results published.  In three of the four cases (TCV, PFBV, and PLPV) we were able to obtain the CP construct in Agrobacterium used by the authors. We used a common bioassay (extension of GFP expression when co-expressed with the silencing suppressor) to reproduce the results in previous papers that show that the TCV, PFBV and PLPV CPs function as silencing suppressors. In one case (MNSV) we reconstructed the CP construct for agroinfiltration and reproduced its silencing suppressor function from a previous publication. Since we validated the silencing suppressor function of these constructs, we do not believe that we need to take the extra step and validate their effect on GFP mRNA levels.  To sum up, we acknowledged the work of others and we reproduced a key experiment from their papers. We should be allowed to build on their studies without having to reproduce the molecular aspects of silencing, especially since our paper is a survey for where silencing suppressors are found within the CPs of the Tombusviridae, and not focused on some aspect on how silencing suppressors function.

Minor points

1.line154: please describe agrobacterium strain. AGL1? 

Response: We have provided a reference for AGL1 (Lazo et al., 1991, Reference 37).

2.Table1: According to Materials and Methods, this experiment was conducted using sap from N. benthamiana infected with TNV, not using virion itself.

Response: We have added a footnote to Table 1 stating that “Virions were present in sap from N. benthamiana leaves.”

3.Figure 4: TNVD indicates TNV?

Response: Yes, we now refer to the TNV strain as TNVDH throughout. 

4.Figure 4: Phylogenetic tree in Scheets et al., 2014 is different from that in Figure 4. Did the authors re-analyze phylogenetic relationship of the CPs?  If so, please describe the method to reconstruct phylogenetic tree in Materials and Methods.

Response: After discussing this point with Dr. Kay Scheets, we realized that we provided the wrong reference for Scheets et al. 2014.  The information used for development of the original phylogenetic tree can be found at < 2014.006b-fP.A.v3.Pelarspovirus.pdf>.  Kay reran the analysis for us and we have now added the methods for the phylogenetic tree into the legend for Fig. 4. She also contributed references for the phylogeny of the tombusvirus CPs.  There are slight differences between the new tree and the original tree, but these differences do not affect the main conclusions of the paper. 

5.Table2: Did the authors check the reaction of Nicotiana species in section Alatae against infection of viruses that used in Table2?

Response:  No, we did not check the response of section Alatae to most of the viruses listed in Table 2.  The structure of our paper was to determine which Nicotiana species responded to TNV virion inoculation with HR. When we found that members of section Alatae responded with HR, we then tested the TNV CP, based on previous experiments with TBSV virions and CP.  Once we found that TNV CP elicited HR, we then considered CPs of other members of the Tombusviridae to be structural variants of a single effector.  At that point it was not necessary to know how Nicotiana species would react to the virus. Our strategy would be similar to direct mutagenesis of a single sequence, only in our case we are utilizing the natural variation that occurs amongst the Tombusviridae.

6.lines, 273-275: The sentence “Most notably, ...” should be moved to discussion section because readers cannot understand what it means without prior information.

Response: we went ahead and deleted this sentence.  MNSV is discussed in detail in the Discussion and we felt that the movement of this sentence to the Discussion would not add to what is already there.

7.lines, 304-319: this section can be omitted or shortened because the strategies and methods to validate elicitor molecules for HR are well-known.

Response:  this section was our response to the reviewer’s main criticism regarding It may not have resonated with this reviewer, but we believe it may provide the rationale for our approach with other readers.

8.lines 374-403/406-414: evolutional and molecular insights on arms race between Nicotiana species and Tombusviridae are a little difficult to follow. I recommend re-organization. 

Response: We have reorganized this section as suggested.  We hope that the revised version will be clearer now.

Round  2

Reviewer 2 Report

I have no more concern with the manuscript except for the following minor corrections.

1) Figure 3: ELISA result should be labeled with C. 

2) Figure 4 or Supplemental Table2: Please describe each accession number.

3) line 373: SS should be spelled out. 

Author Response

Each of the reviewers comments were addressed in the second revision.  Each of these comments addressed issues that clarified the manuscript.  

A "C" was added to figure 3.

Accession numbers were added to Supplemental Table 2.

SS was spelled out to silencing suppressor.

Thank you.